# The Metabolomic Signature of the Placenta in Spontaneous Preterm Birth

**DOI:** 10.3390/ijms21031043

**Published:** 2020-02-04

**Authors:** Summer Elshenawy, Sara E. Pinney, Tami Stuart, Paschalis-Thomas Doulias, Gabriella Zura, Samuel Parry, Michal A. Elovitz, Michael J. Bennett, Amita Bansal, Jerome F. Strauss, Harry Ischiropoulos, Rebecca A. Simmons

**Affiliations:** 1Division of Neonatology, Department of Pediatrics, Children’s Hospital of Philadelphia, Philadelphia, PA 19104, USA; s.elshenawy@gmail.com (S.E.); StuartT@email.chop.edu (T.S.); paschalisdoulias@uoi.gr (P.-T.D.); zurag@email.chop.edu (G.Z.); ischirop@pennmedicine.upenn.edu (H.I.); 2Division of Endocrinology and Diabetes, Department of Pediatrics, Children’s Hospital of Philadelphia, Philadelphia, PA 19104, USA; pinneys@email.chop.edu; 3Center for Research on Reproduction and Women’s Health, Perelman School of Medicine at the University of Pennsylvania, Philadelphia, PA 19104, USA; parry@pennmedicine.upenn.edu (S.P.); elovitz@pennmedicine.upenn.edu (M.A.E.); amita.bansal@hotmail.com (A.B.);; 4Department of Obstetrics and Gynecology, Maternal and Child Health Research Center, Perelman School of Medicine at the University of Pennsylvania, Philadelphia, PA 19104, USA; 5Department of Pathology and Laboratory Medicine, Perelman School of Medicine at the University of Pennsylvania and Children’s Hospital of Philadelphia, Philadelphia, PA 19104, USA; BENNETTMI@email.chop.edu

**Keywords:** placenta, spontaneous preterm birth, metabolism, fatty acids, acylcarnitines

## Abstract

The placenta is metabolically active and supports the growth of the fetus. We hypothesize that deficits in the capacity of the placenta to maintain bioenergetic and metabolic stability during pregnancy may result in spontaneous preterm birth (SPTB). To explore this hypothesis, we performed a nested cased control study of metabolomic signatures in placentas from women with SPTB (<36 weeks gestation) compared to normal pregnancies (≥38 weeks gestation). To control for the effects of gestational age on placenta metabolism, we also studied a subset of metabolites in non-laboring preterm and term Rhesus monkeys. Comprehensive quantification of metabolites demonstrated a significant elevation in the levels of amino acids, prostaglandins, sphingolipids, lysolipids, and acylcarnitines in SPTB placenta compared to term placenta. Additional quantification of placental acylcarnitines by tandem mass spectrometry confirmed the significant elevation in SPTB human, with no significant differences between midgestation and term placenta in Rhesus macaque. Fatty acid oxidation as measured by the flux of ^3^H-palmitate in SPTB placenta was lower than term. Collectively, significant and biologically relevant alterations in the placenta metabolome were identified in SPTB placenta. Altered acylcarnitine levels and fatty acid oxidation suggest that disruption in normal substrate metabolism is associated with SPTB.

## 1. Introduction

Prematurity (delivery before or at 37 weeks gestation) is the leading cause of death in newborn infants, and one in nine babies in the United States are born prematurely [1]. Infants who survive preterm birth often face serious and lifelong health problems, including lung diseases, vision loss, and neurodevelopmental disorders. An Institute of Medicine report in 2007 estimated that premature birth costs the American healthcare system at least USD 26 billion a year [2]. Despite its major impact on infant health, spontaneous preterm birth (SPTB) remains a significant and poorly understood perinatal complication. SPTB includes preterm labor, preterm spontaneous rupture of membranes, preterm premature rupture of membranes and cervical weakness; it does not include indicated preterm delivery for maternal or fetal conditions. While the exact etiology of SPTB remains unknown, there are likely many factors that contribute, including uterine distension, cervical insufficiency, vascular disorders, chorioamnionitis, and placental dysfunction [3,4]. Placental dysfunction is classically associated with fetal growth restriction and preeclampsia; however, emerging evidence supports the concept that placental insufficiency is also associated with a significant proportion of preterm births, especially early preterm births as well as those complicated by chorioamnionitis [5]. Therefore, obtaining a better understanding of factors that contribute to placental dysfunction may lead to a deeper understanding of the etiology of prematurity and open the door for preventative treatment.

During gestation, the placenta supports growth and development of the fetus by facilitating nutrient transport and gas exchange. It produces and releases hormones into both the maternal and fetal circulation to affect uterine function, maternal metabolism, fetal growth, and development. A wide variety of metabolites are produced by the placenta, many of which are involved in energy production [6,7]. The placenta also functions to protect the fetus against xenobiotic molecules, infections, toxins, and maternal diseases [8]. Therefore, a well-functioning placenta is crucial for normal gestation. Healthy term birth is associated with a growing fetus with metabolic demands that gradually begin to exceed the supply capacity of an aging placenta [9]. We hypothesize that deficits in the capacity of the placenta to maintain bioenergetic and metabolic stability throughout the course of pregnancy may ultimately result in SPTB. 

To test this hypothesis, we performed a metabolomic analysis of placenta samples obtained from women with spontaneous preterm deliveries to identify metabolic pathways that may be altered in preterm birth. The aim of our study is to identify novel metabolites and metabolic pathways in the placenta that are altered in the setting of SPTB that will provide insight into the underlying mechanisms driving this process. 

## 2. Results

### 2.1. Clinical Characteristics

Placenta samples for the current study were selected from the larger Cellular Injury and Preterm Birth (CRIB, 821376 NCT02441335) study at the University of Pennsylvania. Placental samples were collected from mid-placenta near the cord insertion on the maternal side and flash-frozen at the time of delivery (within 10 min) and stored at −80 °C prior to metabolite extraction and analysis at Metabolon. The current study utilized a nested case control design with 19 spontaneous preterm (<36 weeks gestation) and control term (between 38 and 41 weeks gestation) sample pairs selected and matched for maternal race (self-identified) and offspring sex from within the larger CRIB study. Twelve spontaneous preterm (SPTB) and 12 control placentas were randomly chosen from this cohort for the metabolomic studies and 7 SPTB and 7 control additional placentas were studied in the validation studies. Due to the matching algorithms used, there were no differences in infant sex, race, or maternal age between term and SPTB placenta samples (Table 1). Of note, none of the preterm or term placental samples were from mothers with preeclampsia or gestational diabetes and none of the women received low dose aspirin for the prevention of preeclampsia. All of the women contributing placenta samples presented in labor with either preterm premature rupture of membranes (PPROM), premature rupture of membranes (PROM), or cervical dilation. All women with preterm labor and none of the women who labored at term received betamethasone treatment prior to delivery. Five of the women with preterm labor received 17-hydroxy progesterone and three received vaginal progesterone for prematurity prevention. None of the women with term delivery received supplemental progesterone. A greater percentage of women with preterm labor received antibiotics, with a primary indication of PPROM or Group B-streptococcus (GBS) prophylaxis; however only 2 of the 19 women with preterm labor were diagnosed with chorioamnionitis. Chronic medications administered to women with preterm labor included albuterol and inhaled corticosteroids for asthma in three women, psychotropic medications in two women, and indomethacin in one woman. Chronic medications documented for women with term labor included albuterol and inhaled corticosteroids in three women, thyroid medication in one woman, and oral acyclovir in one woman.

### 2.2. Global Assessment of Metabolomics Data

Primary metabolomic analysis was performed on placenta tissue from 12 cases with a mean GA of 30 w 5 d and 12 controls with a mean GA of 39 w 5 d. Of the 520 total biochemical compounds detected in the placentas studied, 171 were significantly increased and 17 were significantly decreased in SPTB (*q* ≤ 0.05 vs. term controls; a *q*-value <0.05 was considered significant). Principal component analysis (PCA) identified a shift in the global metabolic profile in placenta of SPTB and found that the two groups were strongly distinguishable (Figure 1). Further, unbiased and supervised classification analysis using the random forest (RF) approach demonstrated that the placental metabolomic profile differentiated the control and SPTB groups with an overall predictive accuracy of 83%, indicating that differences in biochemical profiles between groups were pronounced (Figure 2). 

### 2.3. Amino Acids are Increased in SPTB Placenta

There was an overall increase in amino acid metabolites and peptides in SPTB placenta [10]. Of the 122 metabolites detected in the amino acid pathway, 45 were significantly elevated, while only 3 were reduced in the SPTB placenta (4-imidazoleacetate, indoleacetate and 3-indoxyl sulfate) (*q* ≤ 0.05) (Table 2). Of note, significant changes were seen in the methionine pathway with up to 4-fold changes observed in methionine sulfoxide and N-acetylmethionine sulfoxide, both oxidized forms of methionine, suggesting that there are increased levels of reactive species in SPTB placenta [11]. Levels of acetylated polyamines, N(1)-acetylspermine (1.78-fold elevation, *q* ≤ 0.05), and N-acetylputrescine (3.08-fold elevation, *q* ≤ 0.05) were also increased. Similar elevations in acetylated polyamines, which are markers of oxidative stress were reported by Alexandre-Goubau et al. in very low birth weight placentas [12]. We also observed enhanced ketone bodies oxidation in SPTB placenta. In particular, the increased levels of C4-OH-carnitine may reflect increased alpha-hydroxybutyrate, an organic acid very recently positioned at a crossroad of glutathione biosynthesis and upstream to the TCA cycle through the propionyl-CoA pathway. Taken together, these data show that there is significant oxidative stress in SPTB placenta.

### 2.4. Prostaglandins are Elevated in SPTB Placenta 

Of the 13 metabolites detected in the prostaglandin pathway (monohydroxyl fatty acids and eicosanoids), 10 were significantly elevated (*q* ≤ 0.05) in SPTB placentas while none had decreased concentrations compared to controls (Table 3). We identified marked elevations in PGE2 (2.85-fold change, *q* < 0.05). We also found marked elevations in several oxidized lipids including the eicosanoids, 5-HETE (7.69-fold change, *q* < 0.05), 12-HETE (2.69-fold change, *q* < 0.05), and 15-HETE (3.72-fold change, *q* < 0.05) as well as monohydroxy fatty acids, 9-HODE and 13-HODE (3.09-fold change, *q* < 0.05) (Table 3). While isoprostanes were not measured in this assay, these findings also indicate oxidative stress in SPTB placenta.

### 2.5. Steroid Hormone Levels are Altered in SPTB Placenta

Metabolomic analysis demonstrated an overall reduction in the concentrations of analytes involved in steroid metabolism. Of the 14 metabolites detected in the steroid pathway, levels of 9 metabolites were reduced. Unexpectedly, progesterone was the only steroid hormone that was significantly elevated in SPTB placenta (Table 4). Only 4 of the 12 of the SPTB cases received 17α−hydroxyprogesterone caproate as a prematurity prevention treatment during pregnancy, and none of the controls received this treatment. On further analysis, we found that the mean placenta progesterone concentration was actually lower in cases that received 17α-hydroxyprogesterone caproate than those that did not (1.69 × 10^7^ ± 8.02 × 10^6^ vs. 1.36 × 10^7^ ± 3.01 × 10^6^ ng/dL, respectively), indicating that exogenous progesterone was not likely influencing these results, or masking the extent of decline in progesterone in SPTB. This finding was also unexpected given that placenta progesterone declines with advancing gestation, demonstrating that these changes in SPTB are not related to gestational age.

Estrogen biosynthesis, a cooperative process between the fetal adrenal cortex and the placenta, was also affected, with a reduction in estrone levels as well as the fetal estrogen precursors, DHEA sulfate and 16α-hydroxy-DHEA sulfate. These significant differences may be due to the gestational age at the time of sampling, related to the size of and activity of the fetal adrenal cortex or maternal betamethasone [13]. However, the reported changes in placenta estrogen content that occur as gestation advances are far less than the levels that we report here. The other steroid hormones whose levels were altered in SPTB were androgens and cortisone. The reduction in cortisone levels may be secondary to betamethasone induced alterations in placental 11β-hydroxysteroid dehydrogenase activity as well as an altered NAD:NADH ratio. However, it has been reported that the administration of betamethasone to women before they deliver prematurely did not affect the expression or activity of placental 11-β HSD-2 [14]. The changes in 4-androsten-3β−diol are interesting and could be related to fetal sex and testicular androgen biosynthesis or maternal betamethasone treatment [15]. Taken together, the changes in steroid production are multifactorial, with some reflecting abnormal placenta function in STPB and others resulting from maternal steroid administration.

### 2.6. Sphingolipids and Lysolipids are Increased in SPTB Placenta

There was a significant increase in metabolites from the sphingolipid pathway in SPTB placenta. Of the 24 metabolites that were detected in this pathway, 13 were elevated and no metabolites were decreased (Table 5). Lysolipids were also significantly elevated. Of the 25 metabolites detected in the lysolipid pathway, 21 were elevated in SPTB placenta and none were decreased [10]. 

### 2.7. Acylcarnitines are Elevated in SPTB Placenta

Of the 17 acylcarnitine metabolites detected, 11 were significantly elevated, and none were decreased (*q* ≤ 0.05; Table 6). Given the profound differences in fatty acid and acylcarnitine levels in SPTB and potential implications in both metabolic and inflammatory pathways, we performed additional studies to further quantify acylcarnitine concentrations in SPTB. We employed an assay used in a CLIA-laboratory with stable isotope labeled internal standards to increase specificity, reduce effects of ion suppression, and to allow accurate quantitation for disease to control comparisons. This assay was performed on a separate set of samples within the CRIB cohort to confirm the Metabolon findings. Measurement of acylcarnitine levels were performed on placentas from 9 SPTB cases with a mean GA of 29 w 2 d, and 9 term controls with a mean GA of 39 w 5 d. These placentas were randomly chosen from the original cohort of 19 SPTB and 19 controls. Similar to the data from the global metabolomic analysis, acylcarnitine levels were significantly elevated in the SPTB placenta compared to term control placenta (Table 7). Seventeen of the 41 acylcarnitine metabolites detected were elevated in the SPTB placenta (*p* < 0.02), and none were reduced. Changes were observed predominantly in short chain acylcarnitines but increases in medium and long chain acylcarnitine species were also observed.

Because one of the most significant metabolic pathways that were altered in SPTB placenta was acylcarnitine metabolism, we investigated the possible effect of gestational age on placental acylcarnitine concentrations. Acylcarnitine quantification was performed on gestational age-matched controls of Rhesus monkey placentas at GA of 105 days (comparable to 28-week gestation) and 150 days (full term—40 weeks gestation). In the Rhesus monkey placentas, we found no significant difference between preterm and term placenta but noted a trend towards decreased levels of acylcarnitines in the preterm compared to term placentas (Table 8). These data suggest that the elevation of acylcarnitine species in the human SPTB placentas is not due to a gestational age effect and, therefore, must be a signature of placental dysfunction of preterm birth.

### 2.8. Fatty Acid Oxidation Is Suppressed in SPTB Placenta

In the placenta, acylcarnitines facilitate the transport of fatty acids into the mitochondria where fatty acids undergo β-oxidation to generate acetyl-CoA, NADH, FADH2 and carbon dioxide. Elevated levels of acylcarnitines could reflect a defect in transport or ß-oxidation We quantified the flux of palmitate in placentas from six cases with a mean GA of 24 w 1 d and six controls with a mean GA of 39w 6d. A significantly reduced rate of palmitate oxidation was found in SPTB compared to term placentas (0.14 compared 0.17 pmol/mg/min, *p* = 2.17 × 10^-5^), indicating possible defects in fatty acid oxidation. Over a period of time, reduced rates of fatty acid oxidation will lead to accumulation of upstream metabolites which is reflected in our metabolomic and acylcarnitine quantification results.

## 3. Discussion

We report significant and biologically relevant alterations in the metabolome of SPTB placenta compared to term controls. A major limitation in spontaneous preterm birth research is the lack of human gestational controls. We attempted to address this limitation by investigating acylcarnitine metabolites in Rhesus macaque preterm and term placenta. While this is not a perfect control, our results show that at least for acylcarnitine metabolism, gestational age is not a factor determining differences between SPTB and term placentas. 

Using untargeted metabolomic analysis, we identified multiple pathways that were altered in the SPTB placenta. Levels of sphingolipids, prostaglandins, progesterone, amino acids, and metabolites involved in fatty acid oxidation were significantly different between SPTB and term placenta. Each class of metabolites that were altered may play a role in placental dysfunction through energy failure, inflammation, changes in vasculogenesis, early senescence, and maternal-fetal tolerance. All warrant further investigation to determine whether these changes are causal or merely associated with spontaneous preterm birth. 

Unexpectedly, SPTB placentae had higher levels of progesterone and lower levels of estrogen and estrogen precursors than controls. The biochemical basis for these differences is not known but could reflect alterations in placental and/or fetal adrenal mitochondrial function discussed below that affect cholesterol side-chain cleavage or secretion of steroid hormones. Cholesterol side-chain cleavage, the first committed step in steroidogenesis, takes place in mitochondria. Steroid hormone production is dependent on precursors provided by the fetus and mother to form an integrated fetal–placental–maternal steroidogenic unit [16]. Our observations suggest that the changes in fatty acid oxidation in the SPTB placenta could increase cholesterol side-chain cleavage, thereby increasing progesterone production. The role of placenta estrogens in human SPTB is unclear and remains to be determined.

The most significant changes in SPTB placenta were observed in lipid metabolites. Fatty acids are actively transported into the placenta and are key for normal fetal development [17]. They serve as an energy-yielding substrate and play an important role in fetal energy utilization and placental function. However, when elevated, fatty acids can induce inflammation leading to preterm labor or premature rupture of membranes. Excess inflammation had been proposed as a major cause of SPTB. Of note, was our finding that multiple 2-hydroxy long chain fatty acids were elevated in SPTB placenta. These fatty acids are potent uncouplers of oxidative phosphorylation and have been shown to impair energy homeostasis [18] as well as induce the mitochondria permeability transition pore [19], resulting in significant mitochondrial dysfunction. This finding is in keeping with previous studies showing an altered redox state and oxidative stress in SPTB placenta [20]. 

Acylcarnitine pathways were markedly disrupted in SPTB placenta. Acylcarnitines are intermediate oxidative metabolites consisting of a fatty acyl group esterified to a carnitine moiety that facilitates its transport across the mitochondrial membrane for β-oxidation [21]. In general, acylcarnitines transport long-chain fatty acids into mitochondria and generate acetyl CoA and ultimately ATP but their direct function in the placenta has not been well characterized. Previous studies have demonstrated that the human placenta expresses high levels of enzymes involved in fatty acid oxidation [7]. Accumulating data suggest that acylcarnitines can activate classical proinflammatory signaling pathways, engage pattern recognition receptor (PRR)-associated pathways, and induce the expression of cyclooxygenase-2, resulting in uncontrolled inflammation [22]. Acylcarnitines can also induce mitochondrial dysfunction [21]. This ongoing inflammation and oxidative stress can ultimately lead to severe placental dysfunction, and or disruption of fetal membranes, resulting in preterm birth. Further, elevated circulating levels of free carnitine and several short-chain, medium-chain, and long-chain acylcarnitines have been observed in the setting of adverse pregnancy complications such as gestational diabetes and preeclampsia, both of which are associated with placental dysfunction [21,23].

It has been hypothesized that impaired mitochondrial ß-oxidation is the underlying mechanism leading to accumulation of acylcarnitine species [24,25], which is consistent with our finding of impaired fatty acid oxidation in SPTB placentas. Fatty acid oxidation is a multi-step process in which a fatty acid acyl-CoA is oxidized to yield an acyl-CoA that is 2 carbons shorter, yielding NADH and FADH2 to enter the electron transport chain, as well as an acetyl-CoA molecule that enters the Krebs cycle. Defects in this process can result in the build-up of metabolic intermediaries of the pathways involved. 

Our finding that acylcarnitine metabolites were elevated in SPTB placentas in both the global metabolomic analysis and in a separate set of placenta samples from CRIB that underwent a more sensitive assay for acyl carnitine profiling was a key finding. Furthermore, the lack of acylcarnitine metabolite elevation in the Rhesus monkey placenta suggests that the findings in the SPTB human placentas are not a result of a gestational age effect. However, we acknowledge that Rhesus monkey placenta is not the ideal control, and this is a major limiting factor in our study. Not surprisingly, one of the many struggles of SPTB research is determining whether significant differences between SPTB samples and control samples are due to pathology or are merely the result normal gestation. While some studies have used placentae from medically-indicated preterm births as gestational controls, almost all of these pregnancies in our population are complicated by pre-eclampsia and the metabolic profile of these placentas overlap significantly with SPTB, making their use as controls problematic. Given the obvious difficulty in obtaining preterm gestation age-matched normal placenta samples from humans, we opted to use Rhesus monkey placentas to further evaluate the effect of gestational age on placenta acylcarnitine concentrations as this was the most significantly affected pathway. A study by Eidem et al. compared the transcriptome of human SPTB to term human transcriptome and then overlaid results comparing gestational age-matched controls to term Rhesus macaque placentas after cesarean section without labor [26]. They identified 37 GA specific and 29 SPTB specific candidate genes. While interspecies transcriptome comparison requires careful interpretation, without a better alternative for GA controls, the use of macaque placenta can help to address this major limitation of preterm birth research. We adapted this approach to evaluate our own finding of altered acylcarnitine metabolites. 

A second important finding was the marked elevation of sphingolipids in SPTB placenta. Sphingolipids are a large class of bioactive membrane lipids, and in the placenta, they are critically important in regulating angiogenesis, blood vessel stability, and decidualization [27]. Additionally, their metabolism plays an important role in maintaining maternal–fetal tolerance during pregnancy [27,28]. Animal studies have demonstrated that mice with a defect in sphingosine kinase have increased chemokine production, which promotes neutrophil infiltration and decreased decidual NK cells, leading to feto-maternal intolerance and pregnancy loss [28]. Thus, an altered balance in sphingolipid metabolites may induce placental dysfunction, leading to early labor.

Prostaglandins were also markedly elevated in SPTB compared to term placenta. Prostaglandins are derived from arachidonic acid through a pathway catalyzed by COX2. Activation of the prostaglandin pathway in the fetal placental unit plays a key role in the initiation of labor by increasing sensitivity to oxytocin and triggering myometrial contractions, cervical remodeling, and extracellular matrix degradation through a paracrine effect, leading to rupture of membranes [29]. 

The finding of elevated progesterone in the preterm birth placenta was unexpected since typical levels of progesterone increase with gestation. Of note, 4 of the 12 of the cases received 17-hydroxyprogesterone caproate as a prematurity prevention treatment during pregnancy, and none of the controls received this treatment. 

We observed an overall increase in all classes of amino acid metabolites and peptides in SPTB placenta. Amino acids (AA) are actively transported across the placenta from mother to fetus through a number of well-characterized AA transporters. Impaired mitochondrial metabolism of some amino acids, in particular the branched chain amino acids, can also result in the accumulation of short-chain acylcarnitines. Amino acids are also metabolized in the placenta and the fetus. Thus, while elevated levels of AA in the placenta may be a result of an increase in transport activity, it may also reflect increased catabolism within the placenta. 

Taken together, our data demonstrate that spontaneous preterm birth is associated with significant metabolic changes in the placenta, as reflected in increased catabolism and markers of oxidative stress consistent with early placental senescence. These changes are consistent with placenta insufficiency, which in turn results in an inability to meet the metabolic demands of the fetus. 

The metabolomic analysis of this study is hypothesis-generating, identifying pathways that warrant further interrogation. Additional studies need to be performed to determine if and how these pathways contribute to SPTB. A major strength of our study was the use of an unbiased broad approach to identify significant patterns of metabolic changes in SPTB placenta with adjustment for multiple hypothesis testing. As a result, we have established a rich source of data for future studies examining other pathways that may play a role in spontaneous preterm birth. Furthermore, we performed focused pathway-specific validation of the acylcarnitine changes from the metabolomics analysis. The addition of a functional experiment measuring fatty acid oxidation further validates our findings and provides a mechanism to explain the elevated steady state levels that were observed in SPTB placenta. 

A challenge of studying the placenta is the heterogeneity of the placental structure. We standardized our approach by sampling and preparing the specimens consistently from the maternal side of the placenta. While different cell types have different metabolic signatures, these cell types interact and function as a placental unit, sampling multiple cell types allows us to examine the placenta as a unit.

There are several limitations to this study. One limitation is the lack of human gestational age control, as noted above. While this is a major challenge that exists in spontaneous preterm birth research, previous work by Eidem et al. showed that Rhesus macaque transcriptome can help to disentangle changes associated with GA differences and those associated with spontaneous birth [26]. They classified gene candidates as specific to SPTB or GA. Based on this prior work, we used Rhesus macaque placenta to validate our findings, specifically for alterations in the acylcarnitine pathway, the results of which supported the fact that SPTB, and not GA, leads to these changes. In the study by Eidem et al., they identified ACSL3 as an SPTB specific gene. ACSL3 encodes acyl CoA synthetase long chain family member [23]. The function of this enzyme is to convert free long-chain fatty acids into fatty acyl-CoA esters and thereby plays a key role in lipid biosynthesis and fatty acid degradation.

An additional limitation of our study was the small sample size. We were unable to fully assess the impact of sex of the offspring, among other variables in four cases of IUGR. The increased percentage of IUGR in the cases may reflect underlying placental dysfunction in these cases and may reflect a suboptimal environment with placental dysfunction reflected by poor fetal growth, which is also associated with early labor. In fact, IUGR occurs in a substantial proportion of SPTB, suggesting that the etiology of fetal growth failure and SPTB overlap. Given the major population disparities in preterm birth, larger studies will be necessary to determine whether these findings can be generalized beyond the primarily African American population that was studied. Furthermore, while we evaluated FAO function with palmitate, further studies looking at mitochondrial function and fatty oxidation using other substrates may help characterize the changes in fatty acid oxidation that were observed. SPTB is a broad term that encompasses a variety of etiologies. As we develop better phenotypic classifications, larger studies will also allow for further stratification and can determine whether specific metabolic fingerprints reflect the underlying cause or phenotype of SPTB [3]. 

In conclusion, this study provides an overview of metabolic pathways in the placenta that are disrupted in the setting of SPTB. Many of these pathways have been associated with placental dysfunction in the setting of other diseases of pregnancy, including preeclampsia, intrauterine growth restriction, and gestational diabetes. Growth restriction, preeclampsia, preterm birth, and to an extreme, fetal demise likely reflect a continuum of placental dysfunction. In fact, histological features of placental insufficiency occur in SPTB and are similar to preeclampsia and most cases of fetal growth restriction [30]. How placental failure initiates SPTB remains to be determined, but associated mitochondria dysfunction and oxidative stress result in immune activation which could act on fetal membranes and or the cervix to initiate labor. 

As investigators continue to characterize spontaneous preterm birth, we expect to find heterogeneous phenotypes that all lead to a final common pathway of SPTB. While our metabolic findings may not be unique to SPTB, they do support the idea that a dysfunctional placenta with inadequate compensation cannot sustain a healthy pregnancy. Therefore, placental dysfunction represents an important etiology of spontaneous preterm birth. These findings will lead us to further explore how dysfunctional lipid metabolism through fatty acid oxidation in the placenta plays a major role in metabolic disruption in the setting of preterm birth. Finally, it remains to be determined whether the changes in key metabolic pathways are causal or associated with SPTB. 

## 4. Materials and Methods

### 4.1. Clinical Characteristics

Placenta samples for all of the experiments in the current study were selected from the larger Cellular Injury and Preterm Birth (CRIB 821376, NCT02441335) study at the University of Pennsylvania. CRIB enrollment criteria included women age 18–45 years with singleton pregnancies admitted to the hospital with either spontaneous labor (defined as regular contractions, cervical dilation) or premature rupture of membranes (PROM) occurring between 20 0/7 and 36 6/7 weeks gestational age (pre-term) or at 38 to 41 weeks gestational age (control). CRIB exclusion criteria included multiple gestations, fetal chromosomal abnormalities, major fetal anomalies, intrauterine fetal demise, intrauterine growth restriction, clinical chorioamnionitis, induction of labor, elective cesarean delivery, gestational hypertension or preeclampsia, gestational diabetes. None of the women received low dose aspirin for the prevention of preeclampsia. The current study utilized a nested case control design with 19 preterm and control sample pairs selected and matched for maternal race (self-identified) and offspring sex from within the larger CRIB study (Table 1). Criteria for selection for our study in addition to matching for sex and race was delivery prior to 32 weeks. Thus, only 19 pairs out of the cohort were available for analysis. The CRIB study was approved by the Institutional Review Board at the University of Pennsylvania (protocol #821376) and patients were enrolled after written informed consent.

### 4.2. Metabolomics

Primary metabolomic analysis was performed on placenta tissue from matched 12 cases and 12 controls randomly chosen. There were no differences in patient characteristics between these 12 pairs and the larger cohort of 19 pairs. Placental samples were collected from mid-placenta near the cord insertion on the maternal side and flash-frozen at the time of delivery and stored at −80 °C prior to metabolite extraction and analysis at Metabolon (Durham, NC, USA). The sample preparation process was carried out using the automated MicroLab STAR^®^ system from Hamilton Company. The resulting extract was divided into two fractions: one for analysis by LC/MS/MS and one for analysis by GC/MS. Samples were placed briefly on a TurboVap^®^ (Zymark) to remove the organic solvent. Each sample was then frozen and dried under vacuum. Analytes were extracted and prepared using Metabolon’s standard solvent extraction method [31,32]. The extracted samples were split into equal parts for analysis on complementary GC/MS (gas chromatography mass spectrometry) and LC/MS (liquid chromatography mass spectrometry) platforms. 

The LC/MS portion of the platform was based on a Waters ACQUITY UPLC and a Thermo-Finnigan LTQ-FT mass spectrometer, which had a linear ion-trap (LIT) front end and a Fourier transform ion cyclotron resonance (FT-ICR) mass spectrometer. For ions with signals greater than 2 million, accurate mass measurement was performed. Accurate mass measurements were made on the parent ion as well as fragments. The typical mass error was less than 5 ppm. Ions with less than two million signals required additional efforts to characterize. Fragmentation spectra (MS/MS) were typically generated, but when necessary, targeted MS/MS was employed.

The samples for GC/MS analysis were re-dried under vacuum desiccation for a minimum of 24 h prior to being derivatized under dried nitrogen using N,O-bis(trimethylsilyl)-flouroacetamide (BSTFA). The GC column was 5% phenyl and the temperature ramp ranged from 40 to 300 °C in a 16-min period. Samples were analyzed on a Thermo-Finnigan Trace DSQ fast-scanning single-quadrupole mass spectrometer using electron impact ionization.

Compounds were identified by comparison to Metabolon library entries of purified standards or recurrent unknown entities.

### 4.3. Rhesus Monkey Placentae

To evaluate whether the changes in acylcarnitine metabolites were due to gestational age, placenta samples were collected from non-laboring, mid-gestation pregnancies at 105 days (preterm, *n* = 3) and 150 days (term control, *n* = 3) Rhesus macaques via cesarean delivery. Rhesus macaque placenta is similar in structure to the human placenta; it is a decidual placenta with a villous structure and involves trophoblast invasion for placentation [33]. Samples were flash-frozen and stored at −80 °C (Cincinnati Children’s Hospital Medical Center, University of Cincinnati, Cincinnati, OH, USA) and then sent to the CLIA-certified clinical laboratory at Children’s Hospital of Philadelphia for acylcarnitine profile analysis. All animal procedures conformed to the requirements of the Animal Welfare Act, and protocols were approved prior to implementation by the Institutional Animal Care and Use Committee at the University of California, Davis. 

### 4.4. Acylcarnitine Profiling and Quantification 

To more fully characterize the changes in acylcarnitines identified in the Metabolon analysis, acylcarnitine profiling and quantification were performed on human and Rhesus macaque placenta lysates. To demonstrate reproducibility from the results of the global targeted metabolomics studies, the human samples for this experiment were a unique set of samples of 9 cases and 9 controls from the larger CRIB cohort. Demographic characteristics were identical to the cases and controls used for the Metabolon study. Twenty-five microliters of sonicated placenta were added to a microcentrifuge tube containing pre-dried stable isotopically labeled internal standards (Cambridge Isotope Laboratories, #NSK-B, Tewksbury, MA, USA). Following vortex mixing, 975 µL of ethanol was added and vortex again. Twenty-five microliters of sample were pipetted into a 96-well polypropylene microtiter plate and evaporated to dryness under a stream of nitrogen at 60 degrees C. When dry, 50 µL of butanolic hydrochloric acid (Regis Technologies, Inc., Morton Grove, IL, USA) was added and the plate sealed with foil. The plate was heated to 65 °C for 15 min, followed by drying down over a stream of nitrogen. The dried down samples were reconstituted with 300uL of acetonitrile: water (80:20). Five microliters of each sample were injected into a Xevo TQ-S tandem mass spectrometer (Waters Corporation) and data acquired to collect the parent compounds of mass mz 85. Quantitation was against the nearest chain-length stable isotope labeled internal standard as adapted from the method of Shen et al. [34]. Values were normalized based on protein concentration in the tissue sonicate, which was determined by the Lowry method and the final results reported as µmol/mg protein.

### 4.5. Fatty Acid Oxidation

Fatty acid oxidation rates were measured on placenta samples from the set used for acylcarnitine profiling. Placenta was homogenized into 250 mM Hepes, 1 mM Diethylenetriaminepentaacetic acid (DTPA), 0.1 mM Neocuproine (NC), pH 7.7, plus 1% Triton X-100, and total protein concentration was determined by the BCA assay (Pierce). One mg of homogenate protein was added into 1 mL of Krebs–Ringer bicarbonate solution containing 0.42 mM BSA, 10 mM carnitine, 2.5 mM palmitic acid, and 4 uCi/mL of ^3^H-palmitate and incubated for 2 h at 37 °C. Proteins were precipitated with 10% trichloroacetic acid followed by centrifugation at 8000× *g* for 10 min at 4 °C. The supernatants were neutralized with 10 N NaOH and loaded onto activated AG 1-X8 formate resin chromatography columns. The effluent was collected and the amount of tritiated water measured by scintillation counting. Water is formed during the dehydrogenation of fatty acid in the first and third steps of the cycle. Thus, the formation of tritiated water corresponds to the rate of fatty acid oxidation. Rates were normalized to protein concentrations.

### 4.6. Statistical Analysis

#### 4.6.1. Demographics

Two sample *t*-tests or Mann–Whitney tests were used to compare means between the two groups for continuous variables and Fisher’s exact test was used to compare categorical variables between term and pre-term samples. *p* < 0.05 was considered the threshold for significance for demographic comparisons.

#### 4.6.2. Metabolomics

Following normalization to volume extracted, log transformation, and imputation of missing values, if any, with the minimum observed value for each compound, Welch’s two-sample *t*-test and unpaired Student’s *t*-tests were used to identify compounds that differed significantly between experimental groups [31,32]. For each of the 459 named compounds detected, the data was log transformation and missing values, if detected, were imputed with the minimum observed value for each compound. Welch’s two-sample *t*-tests were used to identify compounds that differed significantly between experimental groups. Correlation analysis was performed comparing relationships between observed changes in metabolite levels and gestational age by ANOVA testing. An estimate of the false discovery rate (*q*-value) was calculated to correct for multiple comparisons that normally occur in metabolomic-based studies and is described as a *q*-value. A *q* value <0.05 was considered significant [35]. Principal component analysis and random forest plots were performed.

#### 4.6.3. Random Forest

Random forest analysis was generated using Array Studio with a script written in R. For a given decision tree, a random subset of the data with identifying true class information was selected to build the tree (“bootstrap sample” or “training set”), and then the remaining data, the out-of-bag (OOB) variables, were passed down the tree to obtain a class prediction for each sample. This process was repeated thousands of times to produce the forest. The final classification of each sample was determined by computing the class prediction frequency (“votes”) for the OOB variables over the whole forest. To determine which variables (biochemicals) make the largest contribution to the classification, a “variable importance” measure was computed and the mean decrease accuracy (MDA) was used as this metric. The MDA was determined by randomly permuting a variable, running the observed values through the trees, and then reassessing the prediction accuracy. If a variable was not important, then this procedure will have little change in the accuracy of the class prediction (permuting random noise will give random noise). By contrast, if a variable was important to the classification, the prediction accuracy will drop after such a permutation, which we record as the MDA. Thus, the random forest analysis provides an “importance” rank ordering of biochemicals; typically, the top 30 compounds in the list as potentially worthy of further investigation are outputted.

#### 4.6.4. Acylcarnitine Quantification

An unpaired Student’s *t*-test was used to determine the statistical significance of differences measured in fatty acid oxidation rates and acylcarnitine levels between SPTB and control placenta. Results presented as significant in these experiments met a threshold of *p* ≤ 0.05.

## Figures and Tables

**Figure 1 ijms-21-01043-f001:**
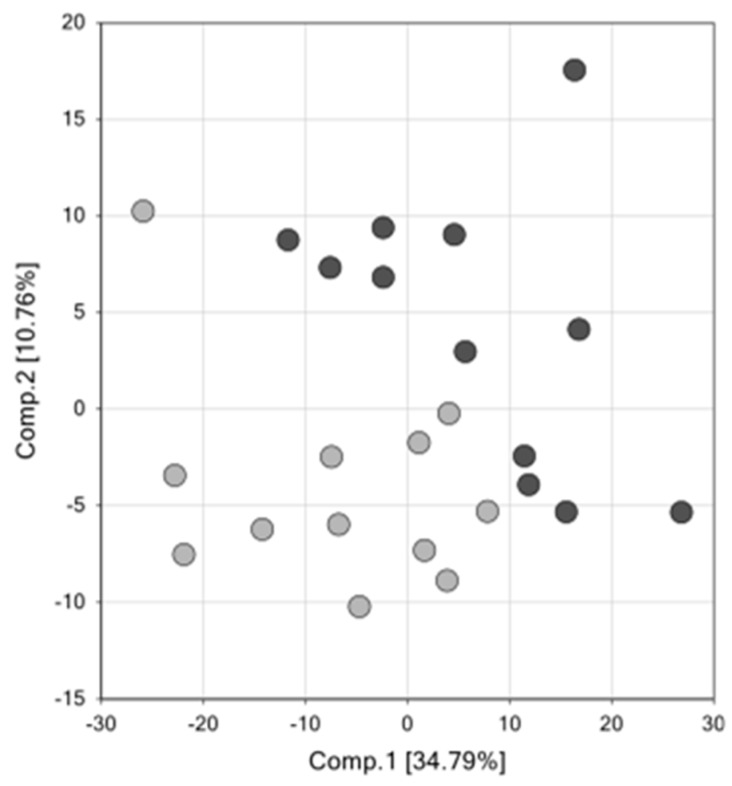
Principal components analysis demonstrates that the global metabolome from placenta samples in spontaneous preterm birth (SPTB) is overall distinguishable from the metabolome of term placenta samples. SPTB placentas: dark gray circles (*n* = 12); Term placenta: light gray circles (*n* = 12). For the analysis shown in Figure 1, the *X*-axis (Comp1) represents 34.79% of the variability between samples and the *Y*-axis (Comp 2) represents 10.79% of the variability between samples.

**Figure 2 ijms-21-01043-f002:**
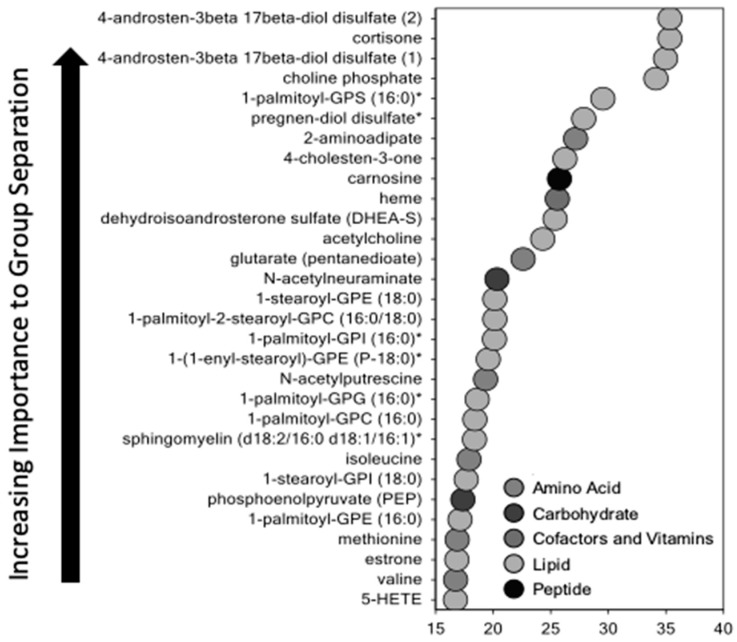
Random forest classification of SPTB placenta samples that underwent metabolomic analysis listed by variable importance, and colored-coded by biochemical class (*n* = 12 preterm and 12 control placenta samples). Random forest classification is an unsupervised analytic tool for ranking the importance of variables within a data set. The *X*-axis represents the mean decrease accuracy (MDA), a calculated metric of biochemical importance in characterizing the overall differences between experimental groups. A higher MDA indicates greater importance in characterizing the differences between pre-term and control placentas.

**Table 1 ijms-21-01043-t001:** Demographics.

	Preterm	Term	*p*-Value
	(*n* = 19)	(*n* = 19)	
Gestational age at birth	28.4 ± 4.5	39.7 ± 0.9	<0.0001
wks, mean ± SD			
Maternal age at delivery	28.3 ± 5.4	26.68 ± 5.0	NS *p* = 0.34
yrs, mean ± SD			
Maternal BMI at first visit	30.0 ± 6.5	27.7 ± 5.2	NS *p* = 0.26
kg/m^2^, mean ± SD			
Race, *n* (%)			NS *p* > 0.99
African American	18 (95)	16 (84)	
White	1 (5)	3 (16)	
Neonatal sex	10 (53)	9 (47)	NS *p* > 0.99
Males, *n* (%)			
Mode of Delivery, *n* (%) *			NS *p* > 0.99
Vaginal	15 (79)	17 (89)	
C-section	4 (21)	2 (11)	
Fetal Growth			
IUGR, *n* (%)	4 (21)	0 (0)	NS *p* > 0.99
Antibiotic administration			
Yes, *n* (%)	11 (58)	4 (21)	NS *p* > 0.99

* All mothers presented in labor.

**Table 2 ijms-21-01043-t002:** Fold change in amino acids that are significantly altered in SPTB placenta.

Pathway	Biochemical Name	Fold Change (Preterm/Term)	*q*-Value
Glycine, Serine and Threonine Metabolism	glycine	1.25	0.0168
	betaine	1.22	0.0473
	betaine aldehyde	2.22	0.0228
	serine	1.26	0.0159
	threonine	1.39	0.0064
	N-acetylthreonine	1.20	0.0469
	N-acetylalanine	1.43	0.007
	aspartate	1.32	0.0227
	asparagine	1.45	0.0057
Glutamate Metabolism	glutamate	1.21	0.0217
	glutamine	1.33	0.0271
	gamma-aminobutyrate (GABA)	1.61	0.0228
	glutamate, gamma-methyl ester	1.87	0.0168
	N-acetylhistidine	1.85	0.007
	imidazole propionate	3.25	0.0152
	4-imidazoleacetate	0.51	0.0457
Lysine Metabolism	lysine	1.24	0.0195
	N-6-acetyllysine	1.45	0.0467
	2-aminoadipate	1.79	0.0007
	glutarate (pentanedioate)	1.77	0.0028
	pipecolate	1.58	0.0278
Phenylalanine and Tyrosine Metabolism	phenylalanine	1.31	0.0167
	tyrosine	1.31	0.0115
	3-(4-hydroxyphenyl) lactate	1.43	0.0207
	N-formylphenylalanine	3.10	0.0064
	indoleacetate	0.22	0.0064
	3-indoxyl sulfate	0.62	0.0261
	C-glycosyltryptophan	1.35	0.0125
Leucine, Isoleucine and Valine Metabolism	leucine	1.30	0.0096
	isoleucine	1.24	0.0114
	3-hydroxy-2-ethylpropionate	1.28	0.0387
	ethylmalonate	1.21	0.0351
	valine	1.36	0.0096
	N-acetylvaline	1.29	0.0228
	isobutyrylcarnitine	2.21	0.007
Methionine, Cytosine, SAM and Taurine Metabolism	methionine	1.34	0.0128
	N-acetylmethionine	1.44	0.0064
	N-formylmethionine	1.27	0.035
	methionine sulfoxide	3.75	0.0136
	N-acetylmethionine sulfoxide	4.52	0.0064
	2-aminobutyrate	1.27	0.0342

Relative values of differentially expressed metabolites in SPTB placenta expressed as fold change of controls. All values are significantly different between preterm and term placenta, *q* ≤ 0.05.

**Table 3 ijms-21-01043-t003:** Fold change in prostaglandins and monohydroxy fatty acids in placenta.

Pathway	Biochemical Name	Fold Change (Preterm/Term)	*q*-Value
Fatty Acid, Monohydroxy	4-hydroxybutyrate (GHB)	1.24	0.0513
	2-hydroxydecanoate	1.14	0.12
	2-hydroxypalmitate	1.55 *	0.0347
	2-hydroxystearate	1.93 *	0.0075
	3-hydroxyhexanoate	1.35 *	0.0342
	3-hydroxyoctanoate	1.47 *	0.0114
	3-hydroxydecanoate	1.33	0.0396
	3-hydroxylaurate	1.63 *	0.0114
	13-HODE + 9-HODE	3.09 *	0.012
Eicosanoid	prostaglandin E2	2.84 *	0.0121
	5-HETE	7.69 *	0.0051
	12-HETE	2.69 *	0.0064
	15-HETE	3.72 *	0.0036

Relative values of differentially expressed metabolites in SPTB placenta expressed as fold change of controls. * significant difference in preterm compared to control, *q* ≤ 0.05.

**Table 4 ijms-21-01043-t004:** Fold change in sterols and steroids in placenta.

Pathway	Biochemical Name	Fold Change (Preterm/Term)	*q*-Value
Sterol	cholesterol	1.86 *	0.0136
	7-alpha-hydroxy-3-oxo-4-cholestenoate (7-Hoca)	0.81	0.0548
	4-cholesten-3-one	1.87 *	0.0016
	7-hydroxycholesterol (alpha or beta)	3.49	0.0666
Steroid	5-alpha-pregnan-3-beta,20-beta-diol monosulfate (1)	1.02	0.1813
	5-alpha-pregnan-3-beta,20-alpha-diol disulfate	0.46 *	0.0064
	5-alpha-pregnan-3(alpha or beta),20-beta-diol disulfate	0.79	0.0638
	pregnen-diol disulfate	0.23	0.0009
	pregnanediol-3-glucuronide	0.77	0.066
	progesterone	1.37 *	0.0347
	cortisone	0.29 *	0.0011
	dehydroisoandrosterone sulfate (DHEA-S)	0.40 *	0.0019
	16-a-hydroxy DHEA 3-sulfate	0.41 *	0.0079
	androsterone sulfate	1.23	0.1259
	4-androsten-3-beta,17beta-diol disulfate (1)	0.09 *	0.0001
	4-androsten-3-beta,17beta-diol disulfate (2)	0.17 *	0.0002
	estrone	0.45 *	0.0064
	pregnanolone/allopregnanolone sulfate	0.64 *	0.0192

Relative values of differentially expressed metabolites in SPTB placenta expressed as fold change of controls. * significant difference in preterm compared to control, *q* ≤ 0.05.

**Table 5 ijms-21-01043-t005:** Fold change in sphingolipids in placenta.

Pathway	Biochemical Name	Fold Change (Preterm/Term)	*q*-Value
Sphingolipid Metabolism	N-palmitoyl-sphinganine (d18:0/16:0)	1.39	0.0592
	sphinganine	0.87	0.152
	phytosphingosine	1.21	0.1485
	palmitoyl sphingomyelin (d18:1/16:0)	1.19 *	0.0036
	stearoyl sphingomyelin (d18:1/18:0)	1.12	0.1251
	sphingomyelin (d18:1/18:1, d18:2/18:0)	1.04	0.1848
	sphingosine	1.18	0.0626
	N-palmitoyl-sphingosine (d18:1/16:0)	1.57 *	0.0038
	sphingomyelin (d18:1/14:0, d16:1/16:0) *	1.31 *	0.0346
	sphingomyelin (d18:2/14:0, d18:1/14:1) *	1.31	0.0565
	sphingomyelin (d18:1/24:1, d18:2/24:0) *	1.34 *	0.009
	sphingomyelin (d18:2/16:0, d18:1/16:1) *	1.39 *	0.0028
	sphingomyelin (d18:1/20:1, d18:2/20:0) *	1.05	0.1751
	behenoyl sphingomyelin (d18:1/22:0) *	1.28 *	0.0263
	sphingomyelin (d18:1/22:1, d18:2/22:0, d16:1/24:1) *	1.41 *	0.0064
	sphingomyelin (d18:1/20:0, d16:1/22:0) *	1.28 *	0.049
	palmitoyl dihydrosphingomyelin (d18:0/16:0) *	1.06	0.1764
	sphingomyelin (d18:1/15:0, d16:1/17:0) *	1.57 *	0.0064
	sphingomyelin (d18:1/21:0, d17:1/22:0, d16:1/23:0) *	1.88 *	0.0064
	sphingomyelin (d18:2/23:0, d18:1/23:1, d17:1/24:1) *	1.98 *	0.008
	sphingomyelin (d18:2/24:1, d18:1/24:2) *	1.28 *	0.0263
	tricosanoyl sphingomyelin (d18:1/23:0) *	1.70 *	0.0036
	sphingomyelin (d18:1/17:0, d17:1/18:0, d19:1/16:0)	1.48 *	0.007
	glycosyl-N-palmitoyl-sphingosine	1.19	0.0929
	lactosyl-N-palmitoyl-sphingosine	1.11	0.2185

Relative values of differentially expressed metabolites in SPTB placenta expressed as fold change of controls. * significant difference in preterm compared to control, *q* ≤ 0.05.

**Table 6 ijms-21-01043-t006:** Fold change in acylcarnitines in placenta.

Pathway	Biochemical Name	Fold Change (Preterm/Term)	*q*-Value
Fatty Acid Metabolism (Acyl Carnitine Species)	butyrylcarnitine	1.33 *	0.018
	propionylcarnitine	1.44 *	0.0311
	methylmalonate (MMA)	1.25	0.0703
	acetylcarnitine	1.31 *	0.0159
	3-hydroxybutyrylcarnitine (1)	1.40	0.0829
	3-hydroxybutyrylcarnitine (2)	1.57 *	0.0067
	hexanoylcarnitine	1.58 *	0.0114
	octanoylcarnitine	1.26	0.0956
	decanoylcarnitine	1.24	0.1164
	cis-4-decenoyl carnitine	1.14	0.2116
	laurylcarnitine	1.92 *	0.007
	myristoylcarnitine	2.16 *	0.0027
	palmitoylcarnitine	2.00 *	0.0019
	palmitoleoylcarnitine	1.83 *	0.0132
	stearoylcarnitine	1.86 *	0.0064
	linoleoylcarnitine	1.60 *	0.049
	oleoylcarnitine	1.72 *	0.0241
	myristoleoylcarnitine	1.58	0.042

Relative values of differentially expressed metabolites in SPTB placenta expressed as fold change of controls. * significant difference in preterm compared to control; *p* < 0.05.

**Table 7 ijms-21-01043-t007:** Changes in acyl carnitine levels found via confirmatory assay in human preterm placenta.

Acylcarnitine Species	Fold Change (Preterm/Term)	*p*-Value
d3-octanoyl(C8) ISTD	1.21	0.38
Acetylcarnitine (C2)	1.59 *	0.01
Propenoyl (C3:1)	1.02	0.94
Propionylcarnitine (C3)	2.39 *	0.01
Butyryl/isobutyryl (C4)	1.65 *	0.01
Tiglyl/methylcrotonyl (C5:1)	1.98 *	0.02
Isovalerylcarnitine (C5)	2.15 *	0.02
3-OH-Butyrylcarnitine (C4-OH)	1.70 *	0.00
Hexanoylcarnitine (C6)	1.52	0.08
3-OH-Isovalerylcarnitine (C5-OH)	2.29 *	0.01
Octenoylcarnitine (C8:1)	1.33	0.14
Octanoylcarnitine (C8)	1.97 *	0.02
Malonylcarnitine (C3-DC)	1.89 *	0.00
Decadienoyl (C10:2)	1.41	0.08
Cis-4-Decenoyl (C10:1)	1.54 *	0.02
Decanoylcarnitine (C10)	1.54 *	0.00
Methylmalonyl (C4-DC)	2.28	0.06
Glutaryl (C5-DC)	2.31 *	0.00
Dodecadienoyl (C12:2)	1.63	0.07
Dodecenoylcarnitine (C12:1)	1.57	0.11
Dodecanoylcarnitine (C12)	1.26	0.09
Adipoyl/Me-glutaryl (C6-DC)	1.21	0.44
Tetradecadienoylcarnitine (C14:2)	1.44	0.06
Tetradecenoyl (C14:1)	1.34	0.09
Tetradecanoyl (C14)	1.49	0.03
Suberyl (C8-DC)	1.90	0.07
3-OH-Dodecenoyl (3-OH-C12:1)	1.47	0.02
3-OH-Dodecanoyl (3-OH-C12)	1.60 *	0.02
3-OH-C14:1	1.68 *	0.02
3-OH-C14	1.88 *	0.01
Hexadecadienoyl (C16:2)	1.33	0.05
Hexadecenoyl (C16:1)	1.33	0.09
Palmitoylcarnitine (C16)	1.39 *	0.01
3-OH-C16	1.50	0.09
Linoleoylcarnitine (C18:2)	1.33	0.14
Oleoylcarnitine (C18:1)	1.19	0.35
Octadecanoyl (C18)	1.46 *	0.02
3-OH-Linoleoylcarnitine (OH-C18:2)	1.36	0.09
3-OH-Oleoylcarnitine (OH-C18:1)	1.61	0.05
C16-Dicarboxylic	1.26	0.25
C18:1-Dicarboxylic	1.36	0.33

Relative values of differentially expressed metabolites in SPTB placenta expressed as fold change of controls. * significant difference in preterm compared to control; *p* < 0.05.

**Table 8 ijms-21-01043-t008:** Fold changes in acylcarnitine Rhesus monkey gestational controls.

Acylcarnitine Species	Fold Change (Preterm/Term)	*p*-Value
d3-octanoyl(C8) ISTD	0.388	0.673
Acetylcarnitine (C2)	0.362	0.826
Propenoyl (C3:1)	0.271	0.662
Propionylcarnitine (C3)	1.081	0.285
Butyryl/isobutyryl (C4)	0.306	0.850
Tiglyl/methylcrotonyl (C5:1)	0.439	0.406
Isovalerylcarnitine (C5)	0.412	0.580
3-OH-Butyrylcarnitine (C4-OH)	0.304	0.760
Hexanoylcarnitine (C6)	0.237	0.363
3-OH-Isovalerylcarnitine (C5-OH)	0.945	0.278
Octenoylcarnitine (C8:1)	0.377	0.759
Octanoylcarnitine (C8)	0.282	0.752
Malonylcarnitine (C3-DC)	0.259	0.448
Decadienoyl (C10:2)	0.309	0.832
Cis-4-Decenoyl (C10:1)	0.377	0.837
Decanoylcarnitine (C10)	0.324	0.949
Methylmalonyl (C4-DC)	0.371	0.665
Glutaryl (C5-DC)	0.309	0.830
Dodecadienoyl (C12:2)	0.300	0.782
Dodecenoylcarnitine (C12:1)	0.301	0.775
Dodecanoylcarnitine (C12)	0.386	0.701
Adipoyl/Me-glutaryl (C6-DC)	0.339	0.977
Tetradecadienoylcarnitine (C14:2)	0.333	0.998
Tetradecenoyl (C14:1)	0.322	0.942
Tetradecanoyl (C14)	0.315	0.880
Suberyl (C8-DC)	0.240	0.187
3-OH-Dodecenoyl (3-OH-C12:1)	0.341	0.904
3-OH-Dodecanoyl (3-OH-C12)	0.357	0.833
3-OH-C14:1	0.297	0.723
3-OH-C14	0.292	0.690
Hexadecadienoyl (C16:2)	0.248	0.608
Hexadecenoyl (C16:1)	0.188	0.323
Palmitoylcarnitine (C16)	0.292	0.714
3-OH-C16	0.306	0.755
Linoleoylcarnitine (C18:2)	0.167	0.485
Oleoylcarnitine (C18:1)	0.148	0.330
Octadecanoyl (C18)	0.222	0.451
3-OH-Linoleoylcarnitine (OH-C18:2)	0.263	0.568
3-OH-Oleoylcarnitine (OH-C18:1)	0.236	0.357
C16-Dicarboxylic	0.310	0.847
C18:1-Dicarboxylic	0.224	0.165

Changes in acyl carnitine levels in Rhesus monkey placenta (*n* = 6) comparing preterm (105 day gestation) to term (150 day gestation). Relative values expressed as fold change of controls. * significant difference in preterm compared to control.

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
