# Peer review of "The Metabolomic Signature of the Placenta in Spontaneous Preterm Birth"

_ijms, 2020, doi:10.3390/ijms21031043_

Round 1

Reviewer 1 Report

Summary:  This paper reports on the cataloging of small molecule metabolites in human placentas obtained from spontaneous preterm births and normal term deliveries.  The work is well-designed and data are presented in tabular form making it easy to understand the data.  The investigators use the appropriate analytical tools for the experiments. 

The work is relevant to the field of perinatal biology and will contribute incrementally to our understanding of the role of metabolism in preterm birth.  However, the study is entirely descriptive and rather than providing a "signature," the study provides a catalogue.  Nonetheless, this information is important to the overall understanding of cellular metabolism in preterm birth.  The investigators acknowledge some shortcomings and the fact that the work is "hypothesis generating." 

I would be helpful and instructive if a summary figure or table would encapsulate the key metabolites in each class that constitute the "signature." 

Author Response

Reviewer 1 suggested that a figure highlighting the major pathways and metabolites that designate SPTB (“signature) would be helpful. We have included a Random Forest Plot which does highlight the most significant pathways and metabolites. If the reviewer and editor want an additional figure, we would be happy to create one.

Reviewer 2 Report

General Notes:

This study is well-designed and provides useful exploratory analysis of placental metabolism. Some parts of the manuscript too strongly contend that the effects are due to prematurity, but I feel the title of the manuscript summarises it well and a signature.

I found the manuscript interesting, and believe others will also, those there are concerns that need to be addressed. These are detailed below.

Abstract:

It is unclear that monkeys are used in the abstract until the Rhesus macaque is mentioned in line 33. You need to clearly state that human and animal studies are performed.

In the abstract it would be worth mentioning gestational ages/cohort/definitions more precisely.

Line 32 – “…significant elevation in SPTB human…” typo?

Line 34 – without specific methods and introduction of methods “flux of 3H-palmitate” is not useful for interpretation. Either clearly state the outcome without refering to the measurement method, or clearly state the methodology and the specific outcome within the abstract.

Line 37 – it is not clear from your results stated why specifically mitochondria might be involved. You need to be clearer on the relation of your results to your state outcomes/interpretation.

Introduction:

Generally well written, and general enough for a non-specialist audience. There are some minor issues with the writing style and definitions that need clarification.

Notes:

You need a definition of preterm birth

Line 43 – “neurodevelopmental disorders” may be a bit better here than “cerebral palsy”, given the audience of the journal, and non-specific descriptions of disorders for other organs (e.g. you state lung disease rather than BPD or CLD, and vision loss rather than ROP).

It is unclear what exactly you mean by spontaneous preterm birth. Do you mean idiopathic or non-induced? Or would something with a clearer cause like chorioamnionitis still be considered SPTB?

Line 57 – 60 “Moreover”, “Furthermore”, “Finally” and “Therefore” are the start of the sentences here. The use of 4 conjunctive adverbs in a row makes for clunky reading, of what are very important, and otherwise clearly stated points.

Given the prominent role of chorioamnionitis in prematurity and placental metabolism, and 2 patients in your sample having this diagnosis this should be mentioned somewhere in the introduction.

Results:

The results are clearly presented and described, which is an achievement for clustered data analysis methods. The inclusion of Rhesus placenta for some of the analysis is commenable, and othercomes a potential major confounder of gestational age effects of the placenta, rather than differences due to prematurity. Please see minor notes below:

Notes:

Was this a registered clinical trial? If not, why not? If so, please include a reference.

It would be beneficial to briefly describe how the placentas were preserved, sampled, and time following birth in this section.

Figure 1 legend states there are light and dark grey circles. This is not the case, please correct.

Figure 2 is not colourblind friendly. (see https://colororacle.org/ or https://www.color-blindness.com/coblis-color-blindness-simulator/). This may be aided by changing colours (or changing to greyscale) and adding figure legend. Also, an arrow on the y-axis title would help indicate the direction of increasing importance of the metabolites.

Have the results of the 2 chorio and 4 IYGR cases been examined compared to rest of the preterm group? A PCA plot of these would be helpful.

Is there a reason the Rhesus placental analysis is limited? You can disentangle the effects of gestation and prematurity in only the acylcarnitine results.

Discussion:

The is a fundamental limitation of this study, and many like it, is that the effects may not be due to preterm birth per se, but rather the differences in placental development due to gestational age. This is mentioned with good detail towards the end of the discussion. However, the beginning of the discussion almost pre-supposes the effects are due to SPTB. Please make the limitations more upfront in the discussion. This will also give an opportunity to discuss a major strength of the stud yearly on, the Rhesus validation.

Though difficult to directly compare, the study would benefit from referring to animal models or other literature which has mapped the metabolic and gestational profile of the placenta over time and compare with findings.

What role do you think betamethasone has on your results (especially steriod metabolism)? Please discuss.

Materials and Methods:

Is there a clinical trial registration?

What is the ethics registration for the Rhesus work?

Author Response

Reviewer 2

Abstract:

It is unclear that monkeys are used in the abstract until the Rhesus macaque is mentioned in line 33. You need to clearly state that human and animal studies are performed. We have added this information.

In the abstract it would be worth mentioning gestational ages/cohort/definitions more precisely. We have added this information.

Line 32 – “…significant elevation in SPTB human…” typo? We have clarified this

Line 34 – without specific methods and introduction of methods “flux of 3H-palmitate” is not useful for interpretation. Either clearly state the outcome without refering to the measurement method, or clearly state the methodology and the specific outcome within the abstract. We have added this information.

Rewritten Line 37 – it is not clear from your results stated why specifically mitochondria might be involved. You need to be clearer on the relation of your results to your state outcomes/interpretation. We have added this information.

Introduction:

You need a definition of preterm birth. We have added this information.

Done Line 43 – “neurodevelopmental disorders” may be a bit better here than “cerebral palsy”, given the audience of the journal, and non-specific descriptions of disorders for other organs (e.g. you state lung disease rather than BPD or CLD, and vision loss rather than ROP). We have changed the wording.

It is unclear what exactly you mean by spontaneous preterm birth. Do you mean idiopathic or non-induced? Or would something with a clearer cause like chorioamnionitis still be considered SPTB? We have changed the wording.

Line 57 – 60 “Moreover”, “Furthermore”, “Finally” and “Therefore” are the start of the sentences here. The use of 4 conjunctive adverbs in a row makes for clunky reading, of what are very important, and otherwise clearly stated points. We have changed the wording.

Given the prominent role of chorioamnionitis in prematurity and placental metabolism, and 2 patients in your sample having this diagnosis this should be mentioned somewhere in the introduction. We have added this information.

Results:

Was this a registered clinical trial? If not, why not? If so, please include a reference. This was a registered trial and thank you for pointing out this omission. We have added the information.

It would be beneficial to briefly describe how the placentas were preserved, sampled, and time following birth in this section. We have added this information.

Corrected Figure 1 legend states there are light and dark grey circles. This is not the case, please correct. We have corrected this and have also changed the colors.

Figure 2 is not colourblind friendly. (see https://colororacle.org/ or https://www.color-blindness.com/coblis-color-blindness-simulator/). This may be aided by changing colours (or changing to greyscale) and adding figure legend. Also, an arrow on the y-axis title would help indicate the direction of increasing importance of the metabolites. We have added an arrow and have also changed the colors.

Have the results of the 2 chorio and 4 IYGR cases been examined compared to rest of the preterm group? A PCA plot of these would be helpful. We do not have the ID  numbers as the study was blinded, but as can be seen by the PCA plot and the original data, there were no outliers.

Is there a reason the Rhesus placental analysis is limited? You can disentangle the effects of gestation and prematurity in only the acylcarnitine results. We would have loved to perform metabolomic studies on the Rhesus monkey samples, but due to cost, we could not.

Discussion:

The is a fundamental limitation of this study, and many like it, is that the effects may not be due to preterm birth per se, but rather the differences in placental development due to gestational age. This is mentioned with good detail towards the end of the discussion. However, the beginning of the discussion almost pre-supposes the effects are due to SPTB. Please make the limitations more upfront in the discussion. This will also give an opportunity to discuss a major strength of the study early on, the Rhesus validation. We have added this discussion to the beginning of the Discussion Section.

Though difficult to directly compare, the study would benefit from referring to animal models or other literature which has mapped the metabolic and gestational profile of the placenta over time and compare with findings. We agree, but we could not find any publications that met this criteria. If the reviewer has a suggested publication, this would be very helpful!

What role do you think betamethasone has on your results (especially steriod metabolism)? Please discuss. We thank the reviewer for pointing out this omission. We have added to the discussion and included additional references.

Materials and Methods:

Is there a clinical trial registration? Yes, and we included this. Thank you for pointing out this omission.

What is the ethics registration for the Rhesus work? We do not have an ethics registration number for animal studies in the United States. We have added that the work was in compliance with the overview committee. Thank you for pointing out this omission.

Round 2

Reviewer 2 Report

I am happy with the revisions, and feel that all comments have been adequately addressed.